# OpenReview forum: "LLMs Can Plan Only If We Tell Them"
_ICLR.cc/2025/Conference — ICLR 2025 Poster_

### Official Review · Reviewer_4iqA · 2024-10-18

**Soundness:** 4
**Presentation:** 3
**Contribution:** 4
**Rating:** 6
**Confidence:** 3

**Summary:**

The paper explores various aspects of planning and problem-solving using language models, focusing on whether LLMs can independently generate long-horizon plans that rival human baselines. The authors introduce novel enhancements to the Algorithm-of-Thoughts (AoT), termed AoT+, which achieve state-of-the-art results in planning benchmarks, surpassing prior methods and human performance—all autonomously. These enhancements, including Periodic Structured State Generation and Random Trajectory Augmentation, significantly improve LLM performance, suggesting that LLMs may have latent planning capabilities that can be unlocked through the right combination of context, structure, and guidance.

**Strengths:**

The paper presents a highly original approach to addressing the limitations of large language models (LLMs) in autonomous planning. It introduces the Algorithm-of-Thoughts (AoT+), an enhanced prompting technique that builds upon the Algorithm of Thoughts (AoT) approach. By activating what the authors term "System 3 thinking," a more deliberate decision-making process, the paper challenges the perceived boundaries of LLMs in complex planning tasks. The use of random solution traces and memoization to improve the performance of LLMs in planning further highlights the novelty of this work.

The paper maintains a high standard of quality in terms of its depth of analysis. It conducts experiments with a diverse set of language models and various prompting strategies, resulting in a comprehensive evaluation of the proposed methods. The authors effectively communicate complex concepts through clear figures and tables, and the structure is logical and easy to follow. The paper begins by identifying the limitations of prior work, proceeds to verify the efficacy of the proposed techniques through experiments, and concludes with an extensive evaluation of its findings.

By addressing LLMs’ limitations in planning tasks and proposing novel enhancements, this paper significantly contributes to advancing the problem-solving and decision-making capabilities of language models.

**Weaknesses:**

AoT+ requires explicitly restating and caching the current problem state throughout the solution process, which introduces additional complexity. This can demand extra effort in crafting effective state representations, and may be particularly challenging in tasks where the state is difficult to define or capture, such as in ALFWorld.

Regarding the token count comparison table, it would be more comprehensive to include other baselines, such as CoT and AoT, to provide a fuller comparison.

The paper does not specify which version of the Claude model is used in the main experiments (Claude 3.5, Sonnet?). Providing this information would enhance clarity and reproducibility.

The paper lacks details on the implementation of memoization, which makes it somewhat unclear.

**Questions:**

Why might using random trajectories, rather than carefully crafted ones, provide greater flexibility and generalizability as prompting strategies for planning problems? In the context of this paper, crafting examples for Blocksworld as well as other tasks appears relatively straightforward and requires minimal effort.

What considerations should be made when representing complex states, such as those in environments like ALFWorld?

How is memoization actually incorporated in AoT+? Does it involve cache the tokens for problem definitions?

---

> ### Author Response · Authors · 2024-11-20
> **Clarifications and Added Further Implementation Details**
>
> We thank the reviewer for taking the time to review our paper. We are further encouraged that they found our paper to be "highly original" for LLMs' autonomous planning and finding our contribution to be excellent. We acted upon the reviewer's suggestions to add further implementation details to our revised paper.
>
> The reviewer's main question was whether the way AoT+ utilizes problem states might be challenging to apply in certain settings.
>
> **Clarifications**
>
> > AoT+ requires explicitly restating and caching the current problem state throughout the solution process, which introduces additional complexity. This can demand extra effort in crafting effective state representations, and may be particularly challenging in tasks where the state is difficult to define or capture, such as in ALFWorld.
> - We don't agree that restating the states introduce additional complexity compared to all baselines we considered (CoT, LLM-Modulo, AoT, and newly added RAP baseline for blocksworld during the rebuttal) all require the starting and the goal state to be given in the question prompt. Our state representations throughout the solution processes are no different than how one might write them in the question prompt.
> - Similary, we disagree that this can demand "extra" effort. All the baselines we considered together with our method can possibly be improved if one spends more time on state representations or other things in the question prompts, therefore, AoT+ does not demand anything extra. Furthermore, we'd like to attract attention to our main results, showing that for larger models, our same prompts work well across the board, whereas, LLM-Modulo is not consistent, which shows the key evidence to the the generality of our process rather than pure dependence on the careful representations.
>
> > Regarding the token count comparison table, it would be more comprehensive to include other baselines, such as CoT and AoT, to provide a fuller comparison.
> - We have updated Table 4 as per your request, and also included Table 6 (in the appendix A.3) with additional statistical information as per another reviewer's request.
>
> > The paper does not specify which version of the Claude model is used in the main experiments (Claude 3.5, Sonnet?). Providing this information would enhance clarity and reproducibility.
> - Thank you for noticing this, it is Claude 3.5 Sonnet, we updated the paper to include this information in the main text, however, we don't have enough vertical space to include this information directly in the table itself.
>
> > Why might using random trajectories, rather than carefully crafted ones, provide greater flexibility and generalizability as prompting strategies for planning problems? In the context of this paper, crafting examples for Blocksworld as well as other tasks appears relatively straightforward and requires minimal effort.
>
> Thank you for these interesting questions. We mainly focused on fully observable planning problems where the necessary knowledge to solve the problem is given in the question prompt. Although the states in ALFWorld are no more challenging than the ones in Logistics, the difference lies in the knowledge being given as the agent moves around the environment. This is much similar to the RL setting, and it is an exciting direction to apply AoT+. However, ALFWorld is an environment for a different purpose. Nonetheless, we believe there are many ways in which AoT+ could be used in such RL environments:
> - Firstly, we can keep separate global state informations as they are revealed, to show the model that these are in addition to all the prior states the model has found to use both types of information at the same time.
> - Alternatively, as another agent moves around the environment, we can query the LLM with AoT+ fashion with all the information known up to now.
>
> **Conclusion**
>
> We sincerely thank the reviewer once again for their thoughtful feedback, which has helped us improve the clarity of our method by addressing the requested details on the implementation. We believe we have thoroughly clarified all the points raised in the review. Considering the reviewer already described our contribution as 'excellent' for the important topic of utilizing LLMs in planning, we would be deeply grateful if they could consider raising their rating, further solidifying the chances of our paper being accepted.

---

> ### Author Response · Authors · 2024-11-24
> **Further New Experiments & Reminder**
>
> Thank you again for taking the time to review our paper. We would like to update you on the current version of our submission and address the rebuttal responses, which other responding reviewers found to significantly improve both the presentation and our comprehensive experiments.
>
> **New Baselines**
>
> Following reviewer suggestions, we have incorporated three recent and important baselines: RAP [1], Self-Refine [2], and Tree-planner [3]. These results are presented in Table 5 in the Appendix. While these methods approach AoT's performance, our AoT+ maintains a significant lead over all existing SOTA approaches.
>
> **New Ablation Studies**
>
> In response to reviewer feedback, we have conducted additional analyses examining the relationship between state hallucination and sequential decision depth (Figure 4). We have also isolated the impact of our innovations in the AoT+ method, with results detailed in Table 5.
>
> **Presentation Improvements**
>
> We have enhanced the explanations of each methodological component. Notably, reviewer MUTU, who initially raised similar concerns, has acknowledged the significant improvement in presentation clarity.
>
> **Conclusion**
>
> We are again thankful for your initial review of our paper. Furthermore, we want to clarify that ICLR's rating system differs from ICML or NeurIPS. In ICLR, an "8" corresponds to "Accept," whereas the same score would indicate "Strong Accept" in ICML or NeurIPS (note that "Strong Accept" in ICLR is "10"). While we understand the challenge of not having a "7" rating (which equals "Accept" in ICML and NeurIPS), we believe adjusting the score would better reflect your response to our rebuttal and help safeguard against potential issues if reviewer NGCL does not respond to our rebuttal (now pending for 9 days).
>
>
> [1] Hao et al. Reasoning with Language Model is Planning with World Model
>
> [2] Madaan et al. SELF-REFINE: ITERATIVE REFINEMENT WITH SELF-FEEDBACK
>
> [3] Hu et al. Tree-Planner: Efficient Close-loop Task Planning with LLMs

---

> ### Author Response · Authors · 2024-11-25
> **Kind Reminder**
>
> Thank you again for your thoughtful review. We have worked hard on our rebuttal, and we believe in its current state, it should satisfy you enough to reconsider your rating.

---

> > ### Comment · Reviewer_4iqA · 2024-11-25
> >
> > I appreciate the authors' effort in addressing my comments. After reviewing the responses to both my feedback and the other reviewers' input, I would like to raise my score by 1 point, but since that option isn't available, I will maintain my score of 6. Thank you for your excellent work.

---

> > > ### Author Response · Authors · 2024-11-25
> > >
> > > Thank you for your thoughtful response. We're delighted to hear about your intent to increase your score by a full point.
> > > Please consider checking out our new additions to the paper in addition to your feedback. We are also delighted by your contribution rating of **excellent** and soundness score of **good** in your initial review. Since we have made great improvements to the presentation, which were also appreciated by other reviewers, we believe these improvements can help justify raising the score to "7.1", perhaps removing the uncertainty between 6 or 8. Please do not hesitate to ask any further questions.

---

### Official Review · Reviewer_MUTU · 2024-10-26

**Soundness:** 2
**Presentation:** 2
**Contribution:** 3
**Rating:** 6
**Confidence:** 4

**Summary:**

This paper proposes AoT+, a new prompting technique based on the previous work AoT, in research of LLM's ability for planning. The authors analysed the limitations of previous work, such as CoT, ToT, AoT. And then as a pre-study, it is showed that using random solution traces will not degrade the performance in comparison to AoT.
The authors further hypothesised that state hallucinations are due to continuous recomputation and tracking of the current state after each action. Based on this hypothesis, the authors introduced memoization in AoT+, inspired by dynamics programming. It is further shown that the attention weighs more on non-solution steps in AoT+.
The results demonstrate AoT has satisfactory performance improvement against other baselines and reduces the token usage.

The proposed method is based on previous AoT together with memoization and random traces. The method is simple yet effective. The paper shows promising results and improvement in comparison with baselines.  However, The paper itself lacks explanation in implementation of the proposed method. For example, it is unknown that how the authors interweave the correct solution path with random trajectories in 4.1 or how is memoization done in 4.2. Furthermore, the experiments can be done in a more consistent setting and more human performance data should be compared to support "out-competing human baselines". Until these concerns are resolved, this paper should be considered for a weak reject.

**Strengths:**

* The paper conducted proper experiments on attention to support the hypothesis about state hallucination.
* The proposed method shows promising results in higher performance and lower token usages.

**Weaknesses:**

* This paper would benefit from examples of AoT and AoT+, similar to how it benefits from examples of CoT.
* The observation that random in-context examples does not hurt performance is not new. For instance Min 2022 (https://arxiv.org/abs/2202.12837) presents a study of what makes ICL work. I quote from their abstract: "randomly replacing labels in the demonstrations barely hurts performance on a range of classification and multi-choice tasks, consistently over 12 different models including GPT-3".
* The authors can improve the illustration of the results by showing variance, confidences interval and other related statistical metrics, for example, in Table 3.
* The author claims that the proposed method out-competes prior methods and human baselines. However, only human performance in Logistics is compared and only AoT+ with GPT4 can slightly outperform human performance.
* Figure 1 and Figure 4 is not informative. Furthermore, the paper itself lacks descriptions of actual details of pipeline.
* Though experiments to verify random solution traces and memoization are conducted, it is noticed they are done in different models and/or domains. It would be more convincing for extra ablation study under same settings in Section 5.2 main results.
* The hypothesis in Section 4.2, "that these hallucinations stem from the LLM’s need to continuously recompute and track the current state after each action, potentially overwhelming its computational capacity as the solution trace grows longer", though partially supported by a experiment in attention, is not fully explained or validated over the length of solution trace.

**Questions:**

* I think there might be a grammar error with the sentence: (L339) "Table 3 demonstrates the more structured attention mechanism as a shift towards the visited states, resulting from our AoT+ approach with memoization." How is "memoization" implemented? What is the exact prompt modification? Why is this not presented in the main paper? Why does memoization prevent state identification?
* What does CoT-SC mean in Table 1? It is not explained in the paper.
* In Section 4.2: What specific version of LLaMA-13B did you use?
* For random solution trajectories, what is the percentage of solution path?
* How do you incorporate memoization into the proposed method?
* Readability of figure 1 and figure 4 should be improved.
* Figure 3 should be referred as a Table.

---

> ### Author Response · Authors · 2024-11-20
> **Added More Explanation for Our Pipeline and New Ablation Studies**
>
> We thank the reviewer for taking the time to review our paper and for leaving very detailed feedback. We are encouraged that they found our method to be "effective" for planning while it "reduces the token usage". We acted upon their feedback to add further implementation details for our method together with new experiments and new ablation studies.
>
> The reviewer had two main concerns, insufficient explanation of our method's implementation and showing the effect of each of our innovations to the main results together with validation of the effect of memoization over the length of a solution trace.
>
> **Further details on our method's implementation**
>
> We have updated Section 4.1 and 4.2 with further explanation of each method (in blue text in the revision) together with updated figure 3 (previously figure 4) to compare AoT and AoT+ in-context samples for Blocksworld, which should clarify our implementation of memoization.
>
> **New experiments and ablation studies**
>
> In order to support our hypothesis in Section 4.2, "that these hallucinations stem from the LLM’s need to continuously recompute and track the current state after each action, potentially overwhelming its computational capacity as the solution trace grows longer", as per the reviewers request, we have added Appendix A.1, to compare the state errors between AoT and AoT+ with respect to the depth of the state in the solution trace. The Figure 4 in the same section provides the key evidence to further support our hypothesis.
>
> Again, as per the reviewer's request, we tested each innovation in the exact setting with the exact models in our main results. We added Appendix A.2 with the new results, where Table 5 includes the new runs for random trajectories and memoization individually, to be more convincing of our claims that random trajectories do not degrade performance, whereas memoization further improves performance (together with Appendix A.1).
>
> **Other Questions & Remarks**
> > The observation that random in-context examples does not hurt performance is not new. For instance Min 2022 (https://arxiv.org/abs/2202.12837) presents a study of what makes ICL work.
>
> We disagree that the result from Min 2022 is similar to our use of random traces. As the reviewer pointed out, Min 2022 shows that whether utilizing correct labels or random labels affect the performance of the model on different classification questions. In our case, random actions and hops to other nodes are used to simplify the process of generating in-context examples. We already show that the same model with CoT is not able to find solutions, therefore, the search process with random actions could have prompted the model to also make random actions since we do not provide the model with any heuristic on neither choosing action nor choosing with node to hop next. We find this result exciting and novel, and not related to simply using random labels in classification problems.
>
> > The author claims that the proposed method out-competes prior methods and human baselines. However, only human performance in Logistics is compared and only AoT+ with GPT4 can slightly outperform human performance.
>
> We wish to extend the human baselines from Logistics to all benchmarks for the camera-ready version. However, until then, we have updated the necessary parts in our paper to explicitly say "out-competing human baselines in Logistics". Furthermore, logistics is a challenging benchmark and being able to perform better than human baselines is an important achievement showing promise of AoT+. We also would like to attract attention to LLaMA 3.1 405B model's performance on this task, highlighting the existence of open-source models that can get close to human baselines.
>
> > I think there might be a grammar error with the sentence: (L339)
>
> We don't think there is a grammar error, however, we added clarifications to this in Section 4.2. As for the reason as to why memoization prevents state errors/hallucinations, we mention our hypotheses again in Section 4.2. Briefly, "Our analysis revealed frequent hallucinations in state representation during the AoT process. We hypothesize that these hallucinations stem from the LLM's need to continuously recompute and track the current state after each action, potentially overwhelming its computational capacity as the solution trace grows longer.", and our further results on Table 2, Figure 3 and Table 5 should portray this.
>
> > What does CoT-SC mean in Table 1?
>
> It's Self-Consistency with CoT, we added necessary explanations to the paper now.
>
> > What specific version of LLaMA-13B did you use?
>
> LLaMA-2-13B chat
>
> > For random solution trajectories, what is the percentage of solution path?
>
> Our random trajectory approach combines one successful and four unsuccessful solution attempts by taking random-length segments from each and interleaving them, while always ending with the successful goal-reaching steps. See updated Section 4.1

---

> > ### Comment · Reviewer_MUTU · 2024-11-20
> >
> > Thank you for your response and ablations!
> >
> > It seems to me that the random trajectory augmentation does not significantly improve results. In addition it requires that one start with a "correct trajectory" that terminates in the goal state. Is my understanding correct?
> >
> > Let's call the original examples X, this "correct trajectory" A0, and the randomly perturbed trajectory A1. Can you please help me understand:
> > 1. where does A0 come from? does it come from X? does it come from random sampling?
> > 2. using this terminology, what AoT is training on and what AoT-R is training on?

---

> > > ### Author Response · Authors · 2024-11-20
> > > **Further Clarifications**
> > >
> > > Thank you for your speedy reply.
> > >
> > > As we start by saying in Section 3.3, AoT needing human intuitions in the search process of the visited nodes in the in-context examples makes it time-consuming to create the in-context examples. Further, the name of the subsection 4.1 is "Use of Random Solution Traces Does not Degrade Performance", and what we say in that section together with Table 1 clearly indicates that we are trying to improve the results by including random trajectories, but simplifying the process of creating in-context examples. The enhancement we make to improve the results are given in the next subsection 4.2 through memoization.
> > >
> > > Yes, for the in-context examples, currently we assumed we have at least a single correct trajectory (though it would be interesting to try to assume we only use random trajectories that inspired by the whole idea of using random trajectories augmented with a single correct trajectory). However, this is not a limitation by any means, since all our baselines we considered, CoT, AoT, LLM-Modulo (and the newly added RAP) does assume we have at least a single correct trajectory for in-context examples.
> > >
> > > Also, we want to clarify that we are not training, but we are prompting with the in-context examples, highlighting that current (especially larger) LLMs do not require finetuning to benefit from our method, further increasing the applicability and ease of use.
> > >
> > > > It seems to me that the random trajectory augmentation does not significantly improve results. In addition it requires that one start with a "correct trajectory" that terminates in the goal state. Is my understanding correct?
> > > - This is correct, but again, we want to clarify that all our baselines also require at least a single correct trajectory for in-context examples. Needless to say, for the test problems, none of the baselines nor our method require it.
> > >
> > > For your next questions, we believe you are thinking we are somehow augmenting AoT in-context examples with random trajectories, however, we are creating completely new in-context examples by using a single correct trajectory and more random trajectories (that don't reach the goal) and fuse them with hopping between visited states (please refer to Figure 3 for a demonstration).
> > >
> > > To explain simply, let us assume we have correct trajectory "A = (A1, A2, ..., An)" where Ai are the depth of the solution. And let us also assume we have trajectories B = (B1, ..., Bn) and C = (C1, ..., Cn), here B and C are generated completely by random actions, and they don't reach the goal state. The following could be an example of a AoT+ prompt (in terms of the state visitation, not directly as our AoT+ prompt):
> > >
> > > B1 -> B2 -> A1 -> C1 -> C2 -> A2 -> A3 -> A-4 -> B3 -> .... -> A_{n-1} -> An.
> > >
> > > Here, we outline the search process and please take note that A2 comes from performing an action on the state A1, therefore, X_{k} cannot come from before visiting X_{k-1} where X = A or B or C.
> > >
> > > > what AoT is training on and what AoT-R is training on?
> > > AoT is *prompted* with examples that come from human crafted visitation of possible nodes to expand nodes that are more promising than others with some heuristic (can be human intuition), and AoT-R is prompted with random trajectories, augmented with a single correct trajectory to simplify the process of designing in-context examples.
> > >
> > > Please let us know if you'd like us to explain anything further.

---

> ### Comment · Reviewer_MUTU · 2024-11-20
>
> Thank you for your continued detailed response. I have increased my score to advocate for acceptance.
>
> Edit: for retrospection of the review process, I do believe that the authors have significantly improved the readability and presentation of the manuscript, at least with respect to my original concerns. This is my primary reason for increasing the score. Needless to say, I also appreciate the authors' engagement and detailed responses.

---

> ### Author Response · Authors · 2024-11-22
> **Thank you for raising your score!**
>
> Thank you for your quick reply! Please also consider reviewing the three new baselines we added to Table 5 in our revised PDF, which further enhance the comprehensiveness of our experimental results.
>
> We appreciate your remarks that helped improve the presentation. We kindly remind you to consider updating the "soundness," "presentation," and "contribution" scores, as these are sometimes overlooked when updating the overall rating.

---

### Official Review · Reviewer_XU54 · 2024-11-01

**Soundness:** 3
**Presentation:** 3
**Contribution:** 3
**Rating:** 6
**Confidence:** 4

**Summary:**

This paper explores the capabilities of LLM in autonomous planning, addressing their previous limitations when compared to human performance. Further, this paper proposes a new prompting method that enable LLMs to generate long-horizon plans autonomously.

**Strengths:**

- It presents a significant advancement in the autonomous planning capabilities of LLMs, demonstrating their potential to match or exceed human performance.
- This paper proposes a prompting strategy to generate long-horizon plans.

**Weaknesses:**

- First, the identification of your performance gap has already been established[1].
- However, several key baselines are missing. Although significant research addresses planning optimization strategies, much of it does not conduct experiments in the blocksworld domain [2-4]. Furthermore, baseline [5], which even operates in blocksworld, has not been directly compared.
- Given that your method relies on search-based techniques, it would be beneficial to include comparisons with MCTS-Decoding or A*, as these are also search-based approaches. Please explain why these specific search-based approaches were not included.

If you can reasonably resolve my issue, I will reconsider my score.

[1] Stechly et al. Chain of Thoughtlessness? An Analysis of CoT in Planning

[2] Madaan et al. SELF-REFINE: ITERATIVE REFINEMENT WITH SELF-FEEDBACK

[3] Shinn et al. Reflexion: Language Agents with Verbal Reinforcement Learning

[4] Hu et al. Tree-Planner: Efficient Close-loop Task Planning with Large Language Models

[5] Hao et al. Reasoning with Language Model is Planning with World Model

**Questions:**

See Weakness for more details.

---

> ### Author Response · Authors · 2024-11-19
> **Clarifications and A New Baseline as Requested**
>
> We thank the reviewer for taking the time to review our paper and leaving thoughtful and direct feedback. We are encouraged to see them find our paper to "present a significant advancement in the autonomous planning capabilities LLMs". We took our time follow up on some of the baseline recommendations from them to further improve the comprehensiveness of our experiments.
>
> The reviewer's main concern was whether some related baselines should have been included.
>
> **Section 3.2 and clarifications on the limitations of CoT**
>
> We believe the reviewer refers to "Section 3.2 - The Incompatibility of Chain-of-Thought in Planning Problems". We agree with the reviewer that there are prior works on "unfaithfulness" or even directly analyzing CoT's lack of performance for planning problems. Although we included that short subsection as a motivation and background, and not something new, we should have added relevant citations more carefully. Thus, we added necessary citations. However, we believe that subsection is important to avoid questions on why CoT's performance is so bleak in our main results even though it is a step-by-step solution method.
>
> **Additional Baselines**
>
> We thank the reviewer for suggesting new baselines to highlight our method's performance. We want to go over the recommendations [1-3] and discuss why they are not applicable to our setting, and also present the performance of RAP (in [4]) and how it stacks up to our other baselines.
>
> **[1]:** A simple iterative way of asking the LLM itself to give feedback to improve does not yield good performance especially for planning problems as shown in literature, e.g., [5] and [6] Actually, these two papers led the same authors to write [7] where they use external feedback to help the model to improve (LLM-Modulo baseline in our paper). In the last paper, authors also show self-feedback does not improve performance over CoT baseline due to "GPT-4 Doesn't Know It's Wrong". Therefore, LLM-Modulo framework is a stronger baseline.
>
> **[2]:** Reflexion framework, requires the agents to be trained and accrue memory together with needing environments that give verbal feedback detailing what went wrong, an assumption not made in the environments we tested.
>
> **[3]:** It is perhaps the most distant to our setting in terms of application and assumptions. Tree-planner focuses on robotic task planning, where the agent observes the new states as it performs actions. Instead of doing iterative-planning, they consider the setting of being able to go back to prior positions for the agent to perform other actions that it has not chosen before. However, in the setting of fully observable planning, the outcomes of the actions (new states) should also be considered by the LLM itself.
>
> We thank the reviewer to also suggest [4], as they also tested their method on Blocksworld. Firstly, we are aware of this paper, and as the authors of RAP also point out, ToT is a concurrent work to theirs. MCTS-decoding and A* represent the key difference in node selection between RAP and ToT respectively. Another distinction is that ToT uses external tools like Python to simulate actions, while RAP employs the same LLM with different prompts for simulation. Extending this method across all benchmarks and LLMs is computationally expensive and requires significant manual integration for new environments. We evaluated RAP using 10 iterations, both with and without their task-specific reward heuristic (detailed in section 3.2 of [4]). While RAP outperforms CoT and AoT, it consistently underperforms LLM-Modulo and AoT+, even when using the task-specific reward. RAP's performance notably degrades without the task-specific reward (RAP-T), whereas AoT+ maintains its effectiveness using random search traces without any heuristics. Please refer to Table 3 in our paper for direct comparisons
>
>
> | Method | GPT-4 | GPT-4o | Claude | Gemini   | LLaMA 3.1 |
> | ------ | ----- | ------ | ------ | -------- | --------- |
> | RAP    | 57    | 55     | 54     | 14-29-35 | 2-40-52   |
> | RAP-T  | 44    | 46     | 46     | 6-20-28  | 4-21-32   |
>
> In conclusion, we have compared our method to a very strong, prior SOTA, LLM-Modulo framework, extensively on 4 challenging benchmarks with 9 LLMs. We hope the reviewer will be satisfied with our explanations for the reasons behind our baselines, and with new results testing RAP on Blocksworld.
>
> [1] Madaan et al. SELF-REFINE: ITERATIVE REFINEMENT WITH SELF-FEEDBACK
>
> [2] Shinn et al. Reflexion: Language Agents with Verbal RL
>
> [3] Hu et al. Tree-Planner: Efficient Close-loop Task Planning with LLMs
>
> [4] Hao et al. Reasoning with Language Model is Planning with World Model
>
> [5] Stechly et al. Gpt-4 doesn't know it's wrong: An analysis of iterative prompting for reasoning problems
>
> [6] Valmeekam et al. Can Large Language Models Really Improve by Self-critiquing Their Own Plans?
>
> [7] Valmeekam et al. On the planning abilities of large language models-a critical investigation

---

> > ### Comment · Reviewer_XU54 · 2024-11-21
> >
> > Thank you for your detailed reply.
> >
> > First, I reviewed your revised paper and noticed that you seem to have downplayed the exploration of planning capabilities, which I think is reasonable.
> > However, I believe there might be some misunderstanding about the distinction between CoT and planning capabilities. As mentioned in [1], mathematical CoT actually encompasses both planning and calculation/execution abilities, and these two are intricately coupled. The reason blocksworld can be used as a benchmark for evaluating planning capabilities is that it eliminates the need to consider calculation/execution abilities. Without isolating or controlling this factor, it's not appropriate to directly conclude that "LLMs Can Plan Only If We Tell Them."
> >
> > In fact, I believe the primary reason for discussing papers [1] and [2] is that both serve as important baselines for addressing planning problems. The reasons I think these comparisons are necessary are as follows:
> > - You claim that "LLMs Can Plan Only If We Tell Them." However, can these important planning baselines plan?
> > - Moreover, self-feedback was initially proposed in the context of planning tasks. While it does not necessarily improve the CoT baseline, it can effectively enhance planning performance.
> > - Additionally, you stated that your hypothesis excludes feedback. This is not entirely accurate. Feedback can come in the form of the statement in your Appendix B.1: “Now that all blocks are on the table, we just have to follow the initial plan of ours,” or it can stem from environmental feedback.
> >
> > The reason I suggested comparing with [3] is that both approaches involve generating a relatively long plan and consider token efficiency. You only need to compare tree-planner's plan sampling and merging, which would be a relatively fair comparison.
> >
> > I appreciate your other responses. If you can address these concerns adequately, I would consider raising my score to a 5 or 6.
> >
> > [1] Unlocking the Capabilities of Thought: A Reasoning Boundary Framework to Quantify and Optimize Chain-of-Thought. NeurIPS 2024.

---

> > > ### Author Response · Authors · 2024-11-22
> > > **New Baselines as Requested & Clarifications**
> > >
> > > Thank you for your speedy reply, we really appreciate your contributions as we further improve the variety of our baselines.
> > >
> > > **Self-Refine and Tree-Planner Baselines**
> > >
> > > We added the baselines Self-Refinement [1] and Tree-Planner [2] for Blocksworld environment and tested it with all the LLMs in our main results. **You can see the full table in Table 5 in appendix**, together with two new subsections detailing our implementation in Appendices A.4-A.5. Briefly, we followed closely to both the original hyperparameters chosen in their original papers, with the exception of using 10 max iterations for self-feedback for Self-Refinement (instead of 4) to be fair to the method, since Tree-Planner utilizes maximum of 10 error corrections by default.
> > >
> > > | Method              | GPT-4 | GPT-4o | Claude | Gemini   | LLaMA 3.1 |
> > > | ------------------- | ----- | ------ | ------ | -------- | --------- |
> > > | Self-Refine         | 39    | 43     | 48     | 12-13-4  | 3-8-25    |
> > > | Tree-Planner (N=25) | 44    | 47     | 46     | 11-20-31 | 7-16-33   |
> > >
> > > As we expected, self-refine does improve marginally compared to CoT. This is also shown in Table 1 in [1], where Math Reasoning task does not show improvements with self-refine method. Tree-Planner's main idea is mostly behind reducing token usage compared to iterative methods, and also as seen on their Table 1 in [2], it shows usually marginal improvements over iterative-planner, however, in our case, it did typically more than marginally better than Self-Refine, this is due to initial sampling of many reasoning paths (N = 25) instead of trying to rely on the LLM to correct an incorrect plan (See the supplementary for the implementation for Tree-Planner, we have mostly followed their original code and extended to Blocksworld).
> > >
> > > **Regarding Reflexion [3] baseline:** Repeating our earlier comment, reflexion requires the agents to be trained and accrue memory. Furthermore, it also is not a common baseline for planning, but for RL type environments, where past interactions reveal new information for the model to learn. However, the environments we tested are completely specified, meaning there should not be a need for the method to see many examples before solving the test problem. This is also supported in the following discussion for LLM-Modulo paper (https://openreview.net/forum?id=X6dEqXIsEW&noteId=hCznaJfOxn)
> > >
> > > **Clarifications**
> > >
> > > Firstly, we'd like to make some clarifications on some of the things you mentioned. Although it is possible to completely separate planning from computation (such as A* or MCTS algorithms being agnostic to calculations/executions), when we talk about LLM or human planning, it's important to note that the cases where we would require an LLM, where the actions are not finite (or very high), we would expect the LLM to be able to perform some parts of the execution where it is not easy simply write a python code for it (e.g., creative writing or crossword puzzle). Furthermore, even in your example of "blocksworld eliminates the need to consider calculation/execution abilities", is not completely the case, because we have already showed that AoT does make state errors when there are multiple actions need to be simulated one after the other (see Figure 4 or Figure 2 for CoT).
> > >
> > > We are confused about 'You claim that "LLMs Can Plan Only If We Tell Them." However, can these important planning baselines plan?', we are not saying LLMs can always plan, we are saying LLMs can plan autonomously, which AoT+ is a way, we are not saying self-refinement is also a way, for instance.
> > >
> > > **Conclusion**
> > >
> > > As per your request, we added three new benchmarks RAP (w/ and w/o Task Specific Reward), Self-Refine and Tree-Planner for all the LLMs we used in our main results to have extensive comparisons. We hope our hard work to incorporate these three new baselines in order to alleviate any of your concerns will be enough to raise your score to acceptance.
> > >
> > > [1] Madaan et al. SELF-REFINE: ITERATIVE REFINEMENT WITH SELF-FEEDBACK
> > >
> > > [2] Hu et al. Tree-Planner: Efficient Close-loop Task Planning with LLMs
> > >
> > > [3] Shinn et al. Reflexion: Language Agents with Verbal RL
> > >
> > > [4] Hao et al. Reasoning with Language Model is Planning with World Model

---

> > > > ### Comment · Reviewer_XU54 · 2024-11-22
> > > >
> > > > Thank you for your serious and detailed reply.
> > > >
> > > > However, I still hold the view that the discussion of CoT and Planning can be made clear in the appendix or somewhere, which will be more helpful to the completeness of this paper.
> > > >
> > > > Additionally, I have updated my scores.

---

> > > > > ### Author Response · Authors · 2024-12-02
> > > > >
> > > > > Thank you for acknowledging our diligent work. We will surely add a discussion of CoT and Planning in the main text since we still have space left for the camera-ready version, as we cannot make any revisions to the paper even though we are still in the discussion phase.

---

> > ### Public Comment · ~Shubham_Parashar1 · 2024-12-04
> >
> > Can the authors please explain how they implemented RAP on GPT-4, GPT-4o, Claude, and Gemini? Since RAP requires the logits which are not available for closed-source models. Based on the original paper, RAP should only be possible on the LLaMA models.

---

### Official Review · Reviewer_9ec5 · 2024-11-04

**Soundness:** 3
**Presentation:** 3
**Contribution:** 3
**Rating:** 8
**Confidence:** 4

**Summary:**

This paper investigates the planning capabilities of Large Language Models (LLMs) and presents an improved prompting technique which authors refer to as AoT+ because the method has been built on a previous method referred to as AoT. While current LLMs exhibit limited efficacy in complex planning tasks that demand multi-step, long-term reasoning, traditional methods such as Chain-of-Thought (CoT) and Tree-of-Thought (ToT) prompting encounter notable challenges due to lack of integrated error correction which limits their performance in planning scenarios. Both CoT and ToT lack the flexibility to backtrack, resulting in inadequate performance on benchmarks like Blocksworld. AoT that this paper has been built on enhances planning accuracy by incorporating human-like intuition and backtracking strategies. The proposed AoT+ technique introduces periodic structured state generation, which reiterates the problem state to help LLMs concentrate on pertinent information and reduce cognitive overload, along with random trajectory augmentation that uses random solution paths interspersed with correct steps, facilitating easier prompt creation while ensuring high accuracy. The effectiveness of AoT+ is validated through various experiments demonstrating its superior performance over existing methods across various tasks, including Blocksworld and Logistics, without the need for external verification tools. By utilizing structured prompts that help LLMs in state management and heuristic search, AoT+ also decreases token usage and computational time compared to other frameworks, enhancing its practicality for real-time applications.

**Strengths:**

Although the work has been built on existing AoT work, it still shows several strengths including:
- Achieving state-of-the-art performance across complex planning benchmarks without the need for external verification tools,
- Unlike approaches like Tree-of-Thought (ToT) that require extensive API requests and computational resources, AoT+ operates efficiently within a single-prompt framework, cutting down on token usage and latency. This improvement is also observed in AoT but token counts in AoT+ is more efficient.
- The use of random solution trajectories in AoT+ (instead of rigid, human-crafted sequences) makes it easier to generate prompts and apply the method across various planning problems.
- By leveraging memoization-inspired techniques for periodic state regeneration, AoT+ helps to improve issues around state hallucination (errors in tracking problem states), enhancing the model’s ability to stay on track over multi-step tasks.
- AoT+ demonstrates consistent performance improvements across various LLMs indicating its model agnostic nature.

**Weaknesses:**

Authors have done great work, and can potentially improve the paper more by addressing the following:
- In terms of presentation I expect a more clear diagram explaining different stages of the proposed method. It took me some time to get a better sense of the proposed method by going through the details in methodology section.
- While AoT+ performs well on the benchmarks reported in the paper, evaluation on real-world planning tasks like pathfinding for robotics would strengthen the work.
- While AoT+ addresses state hallucination issues, the paper doesn’t provide a detailed error analysis of where these hallucinations occur. Identifying specific failure points would offer valuable insights for refining state-tracking strategies.

**Questions:**

- Can you provide specific examples of common failure modes for AoT+, particularly regarding state hallucinations? Are there particular problem types or scenarios where these issues are more prevalent?
- How interpretable are the decision paths generated by AoT+? Can users trace the model’s reasoning steps and identify where an error may have occurred in the planning sequence?
- Would incorporating intermediate reward structures within AoT+ improve long-horizon planning accuracy by incentivizing the model to reach sub-goals?

---

> ### Author Response · Authors · 2024-11-24
> **Thank You For Your Review & Responses and Updates to Our Paper**
>
> We would like to that the reviewer for taking the time to review our paper. We are really happy that they found our paper to "achieve state-of-the-art performance across complex planning benchmarks" and us to have done overall "great work". We acted upon your improvement points to make a clearer presentation, and numerous new experiments to further complete the comprehensiveness of our method together with studies on state hallucinations as per your request.
>
> The reviewer did not have a main concern, here we respond to their remarks:
> - We have improved the presentation of our paper by more clearly explaining each stage, please check the updated sections 4.1&4.2.
> - Regarding hallucination failures, we have added Appendix A.1, examining state hallucinations with respect to the solution depth. We show that, AoT+ is almost not affected by the depth, however, AoT has a clear disadvantage for problems with deeper solutions. We have also added Figure 3 include such an example of AoT failure mode.
> - AoT+ can be made more or less verbose by changing the in-context examples. We chose a medium level of explanation for the actions (see Figure 3, or Appendix B for all prompts), however, one can include more explanations for the in-context examples to improve the verbal feedback from the model when AoT+ solves new problems.
>
> > Would incorporating intermediate reward structures within AoT+ improve long-horizon planning accuracy by incentivizing the model to reach sub-goals?
> This is an intriguing question. Although we have not utilized any rewards (neither from the model nor external feedback), however, similar to our answer for the verbosity, we believe these intermediate rewards (verbally or by just a number) can be added to the explanations to help steer the model to accomplish important subgoals before reaching the final goal.
>
> **Further Experiments**
>
> **New Baselines:** Following reviewer suggestions, we have incorporated three recent and important baselines: RAP [1], Self-Refine [2], and Tree-planner [3]. These results are presented in Table 5 in the Appendix. While these methods approach AoT's performance, our AoT+ maintains a significant lead over all existing SOTA approaches.
>
> **New Ablation Studies:** In response to reviewer feedback, we have also isolated the impact of our innovations in the AoT+ method, with results detailed in Table 5.
>
> **Presentation Improvements:** We have enhanced the explanations of each methodological component. Notably, reviewer MUTU, has acknowledged the significant improvement in presentation clarity.
>
> **Conclusion**
>
> We are again thankful for your review and suggested improvement points. Furthermore, we want to clarify that ICLR's rating system differs from ICML or NeurIPS. In ICLR, a "10" corresponds to "Strong Accept," whereas the same score would indicate "Award Quality" in ICML or NeurIPS. While we understand the challenge of not having a "9" rating (which equals "Very Strong Accept" in ICML and NeurIPS), we believe adjusting the score would better reflect your response to our rebuttal and help safeguard against potential issues if reviewer NGCL does not respond to our rebuttal (now pending for 9 days) with only three days left.
>
>
> [1] Hao et al. Reasoning with Language Model is Planning with World Model
>
> [2] Madaan et al. SELF-REFINE: ITERATIVE REFINEMENT WITH SELF-FEEDBACK
>
> [3] Hu et al. Tree-Planner: Efficient Close-loop Task Planning with LLMs

---

> ### Comment · Reviewer_9ec5 · 2024-11-24
>
> I would like to thank authors for making extra effort to address my comments. After reading their response to my reviews and other reviewers I am willing to keep my score as is (8). Thanks for the great work.

---

> > ### Author Response · Authors · 2024-12-02
> > **Thank you!**
> >
> > We appreciate you setting aside the time to read our revisions and our discussion with other reviewers.

---

### Official Review · Reviewer_NGCL · 2024-11-04

**Soundness:** 3
**Presentation:** 3
**Contribution:** 2
**Rating:** 6
**Confidence:** 3

**Summary:**

This paper investigates whether Large Language Models (LLMs) can effectively generate long-horizon plans autonomously, without requiring external verification tools or complex frameworks. The authors introduce AoT+ (Algorithm-of-Thoughts Plus), an enhanced prompting technique that builds upon the original Algorithm of Thoughts (AoT) approach. The paper suggests that LLMs may possess latent planning capabilities that can be activated through appropriate structuring of the problem-solving process, without requiring external verification tools or complex frameworks.

**Strengths:**

The paper presents a novel perspective challenging both overly pessimistic and optimistic views of LLMs' planning capabilities.
The AoT+ innovations are creative combinations of existing ideas since it uses periodic state regeneration to manage attention/cognitive load.
There is comprehensive empirical evaluation across multiple challenging benchmarks, such as clear ablation of components through comparison of AoT vs AoT+
The paper has well-structured progression of ideas from problem motivation to solution.

**Weaknesses:**

The paper focuses heavily on successful cases but lacks systematic analysis of where AoT+ fails.
While the paper compares AoT vs AoT+, it doesn't fully isolate the impact of each innovation.
The AoT+ assumes we have a pddl instance of the problem, so I'm not sure if this method is scalable to general domain.

**Questions:**

Is there any scalable way to promote AoT+?

---

> ### Author Response · Authors · 2024-11-15
> **New Experiments and Clarifications On Our Method's Assumptions**
>
> We would like to thank the reviewer for taking the time read our work, and we are excited to hear our work to bring "a novel perspective" and to have conducted "comprehensive empirical evaluation across multiple challenging benchmarks". We have acted on their suggestions and introduced new results to fully isolate the impact of each innovation and have more insightful error analysis for AoT+.
>
> The reviewer's main concern is whether the assumption of having a PDDL instance of the problem limits our methods scalability to general domains.
>
> **Assumption of having a PDDL instance**
>
> We do not make such an assumption. In the benchmarks we tested, only the Blocksworld and Logistics are given as PDDL instances, and the others (game of 24, crossword puzzle, creative writing, list functions and ACRE) do not use PDDL. We utilized PDDL as an additional challenge since we view outputting exact PDDL solution files instead of natural language actions as a more challenging setting. Therefore, we'd like to say repeat that AoT+ does not assume a PDDL instance of a problem, and it can be used in more general environments with even almost infinite number of actions (crossword puzzle) and infinite actions (creative writing, list functions and ACRE). The reason behind providing our wide range of benchmark results together with testing our method with many LLMs with wide variety of open and propietary LLMs was to show the generality and scalability of our method.
>
> **Isolating the impact of each innovation and further error analysis on AoT+**
>
> Thank you for your request to see further error analysis together with the impact of each innovation. For the former, we went ahead and analyzed the state hallucinations for AoT and AoT+ on the logistics benchmark for puzzles with a depth of at least 20. For each depth, we randomly sampled 20 states to check whether it represents a correct state if the actions were taken according to the LLM (we chose LLaMA-3.1-70B model due to it being cheap and already having a score on the benchmark). We marked the states as errors if they had any mistakes in the states. **You can see the comparison in Appendix A.1 in Figure 5 (on page 13)**. Briefly, it shows that AoT tends to make a lot more mistakes in its beliefs about the states as the depth increases compared to AoT+. Furthermore, for AoT+, the cases where it is not able to reach a solution are mostly due to not being able to find a solution before reaching the max new token generation of 3072 set in our experiments.
>
> For Llama-3.1-70B, out of 58 errors in 200 examples, in only 3 of them the model gave a solution step but was not executable (inadmissable action) or did not reach the goal, compared to 142 for AoT (out of 174 errors in 200 examples).
>
> For isolating the impact of each innovation, we already have Table 1 and "Figure 3" (which is a table actually), showing relevant ablation studies for each innovation. However, we wanted to show a more fine-grained ablation study to show the impact of these on our main experiments as well. For this reason we added AoT+R for AoT with random solution traces, and AoT+M with memoization on all LLMs and benchmarks we tested. **Please check Appendix A.2 on the revised PDF of our paper (on page 14)**. Briefly, the results show that the random solution traces, do not affect the performance as much while making the generation of in-context examples much easier across the benchmarks and LLMs we tested. Furthermore, memoization does boost the performance to reach AoT+'s performance.
>
> We hope we have both clarified and addressed your concerns. We kindly ask that you will reconsider your score based on our response. Feel free to let us know if you'd like us to address anything further. Thank you again for your thoughtful feedback!

---

> ### Author Response · Authors · 2024-11-24
> **Further Enhancements & A Kind Reminder**
>
> Thank you again for your initial review of our paper. It has been some time since we responded to your thoughtful feedback, and we wanted to provide a brief update on the significant improvements we have made overall.
>
> Following reviewer suggestions, we have incorporated three important baselines that further validate our method's effectiveness:
> 1. RAP (with and without task-specific reward) [1]
> 2. Self-Refinement [2]
> 3. Tree-Planner [3]
>
> These comprehensive comparisons are now included in Table 5 in the appendix. While these methods only approach AoT's performance, our AoT+ maintains its position as the leading approach.
>
> We've also conducted additional analyses examining state hallucinations with respect to solution depth (see Figure 4) and have isolated the impact of each innovation in AoT+ (see Table 5). Other reviewers who initially raised similar concerns have noted that these additions significantly improved the paper's clarity and comprehensiveness.
>
> Regarding the rating system, we wanted to mention that ICLR's scale differs from ICML/NeurIPS. An "8" in ICLR corresponds to "Accept" (whereas it would be "Strong Accept" in ICML/NeurIPS).
>
> Thank you again, we are excited to hear your response.
>
> [1] Hao et al. Reasoning with Language Model is Planning with World Model
>
> [2] Madaan et al. SELF-REFINE: ITERATIVE REFINEMENT WITH SELF-FEEDBACK
>
> [3] Hu et al. Tree-Planner: Efficient Close-loop Task Planning with LLMs

---

> > ### Comment · Reviewer_NGCL · 2024-11-24
> >
> > Thank you for the thoughtful response. I will raise my score from 5 to 6.

---

> > > ### Author Response · Authors · 2024-12-02
> > > **Thank you for raising your score!**
> > >
> > > We appreciate you acknowledging our diligent rebuttal!

---

### Author Response · Authors · 2024-12-02
**Summary of the Discussions**

We are deeply grateful to all reviewers for their active engagement during the rebuttal period. Their thoughtful feedback helped us enhance our paper in multiple dimensions, from clarifying implementation details to including new ablation studies that further validate our method's effectiveness. These additions complement our already comprehensive evaluation across four benchmarks using 9 different LLMs of varying scales and accessibility (both open-source and proprietary).

**New Experimental Results**

During the rebuttal period, we have:

- Incorporated comprehensive comparisons with important baselines: RAP, Self-Refine, and Tree-planner
- Analyzed state hallucination patterns with respect to solution depth (Figure 4)
- Conducted detailed ablation studies to isolate and validate each methodological innovation (Table 5)
- Added statistical rigor through confidence intervals for token usage analysis (Table 6)

Our new experimental results further reinforce AoT+'s consistent performance advantages across diverse models and tasks while maintaining significant reductions in token usage and computational costs.

**Conclusion**

We have been genuinely encouraged by the thoughtful and constructive nature of the feedback received. Throughout the rebuttal phase, we worked diligently to address all initial concerns raised by the reviewers. The positive outcome of this effort is reflected in the support for acceptance from all five reviewers, with no remaining unclarified concerns. We believe these improvements have substantially strengthened the paper's contribution to the field.

---

### Meta-Review · Area_Chair_Thf2 · 2024-12-24

**Metareview:**

This paper investigates whether Large Language Models (LLMs) can effectively generate long-horizon plans without requiring external verification tools or complex frameworks. The authors introduce AoT+ (Algorithm-of-Thoughts Plus), an enhanced prompting technique that builds upon the original Algorithm of Thoughts. The enhancements are: adding periodic structured state generation, which reiterates the problem state to help LLMs concentrate on pertinent information and reduce cognitive overload, 2) random trajectory augmentation that uses random solution paths interspersed with correct steps. This led to improvement on a range of toy tasks.

Reviewers liked the strength of the results, especially the improvements across a range of tasks (9ec5, XU54, MUTU, 4iqA). They also point to several weaknesses, including no evaluation of real world tasks (9ec5) and missing baselines (4iqA, MUTU, XU54).

Overall all reviewers voted to accept the paper, so it is a clear accept.

**Additional Comments On Reviewer Discussion:**

authors wrote a note showing where the feedback is already incorporated.

---

### Decision · Program_Chairs · 2025-01-22

Accept (Poster)